# Test-Time Adapted Reinforcement Learning with Action Entropy Regularization

**Shoukai Xu** [* 1] **Zihao Lian** [* 2] **Mingkui Tan** [1 3 4] **Liu Liu** [2] **Zhong Zhang** [2] **Peilin Zhao** [2 5]

## Abstract

Offline reinforcement learning is widely applied in multiple fields due to its advantages in efficiency and risk control. However, a major problem it faces is the distribution shift between offline datasets and online environments. This mismatch leads to out-of-distribution (OOD) state-action pairs that fall outside the scope of the training data. Therefore, existing conservative training policies may not provide reliable decisions when the test environment deviates greatly from the offline dataset. In this paper, we propose Test-time Adapted Reinforcement Learning (TARL) to address this problem. TARL constructs unsupervised test-time optimization objectives for discrete and continuous control tasks, using test data without depending on environmental rewards. In discrete control tasks, it minimizes the entropy of predicted action probabilities to decrease uncertainty and avoid OOD state-action pairs. For continuous control tasks, it represents and minimizes action uncertainty based on the normal distribution of policy network outputs. Moreover, to prevent model bias caused by overfitting and error accumulation during the test-time update process, TARL enforces a KL divergence constraint between the fine-tuned policy and the original policy. For efficiency, TARL only updates the layer normalization layer parameters during testing. Extensive experiments on popular Atari game benchmarks and the D4RL dataset demonstrate the superiority of our method. Our method achieved a significant improvement over CQL, with a 13.6% episode return relative increase on the hopper-expert-v2 task.

*Equal contribution. This work is done when Zihao Lian works as an intern in Tencent AI Lab. [1]South China University of Technology [2]Tencent AI Lab [3]Pazhou Laboratory [4]Key Laboratory of Big Data and Intelligent Robot, Ministry of Education [5]Shanghai Jiao Tong University. Correspondence to: Peilin Zhao <masonzhao@tencent.com>, Mingkui Tan <mingkuitan@scut.edu.cn>.

*Proceedings of the 42nd International Conference on Machine Learning*, Vancouver, Canada. PMLR 267, 2025. Copyright 2025 by the author(s).

## 1. Introduction

Reinforcement learning is an important research field in artificial intelligence and has achieved remarkable success. Offline reinforcement learning (Kumar et al., 2020; An et al., 2021; Fujimoto & Gu, 2021; Kostrikov et al., 2021) offers significant advantages, enables efficient data utilization and low risks by learning promising policies from pre-collected datasets. Therefore, offline reinforcement learning has become an indispensable method for a wide range of applications. For example, offline reinforcement learning can learn an optimal driving policy from a large driving dataset to improve the safety and stability of autonomous driving (Yu et al., 2020a; Liu et al., 2023). Furthermore, offline reinforcement learning can also be applied to robotics control (Kalashnikov et al., 2018), game AI (AlphaStar; Ye et al., 2020), portfolio selection (Zhang et al., 2023; Xu et al., 2020), recommendation systems (Li et al., 2010; Gao et al., 2022) and other fields, where it can improve the intelligence of robots or agents by learning from large amounts of offline data.

The dataset used in offline reinforcement learning can be collected through interaction with the environment based on a fixed policy, simulation runs or human demonstrations. The state-action pairs in the offline dataset are unable to cover all the possibilities of the state-action space. Consequently, the dataset inevitably exhibits a certain degree of distribution shift from the true state-action visitation frequency associated with the learned policy, as shown in Figure 1. During the deployment process, the policy may encounter novel state-action pairs, which subsequently leads to the predicament of being incapable of coping with unforeseen changes and dynamic environments in practical applications. Specifically, offline reinforcement learning may face out-of-distribution (OOD) samples, *i.e.,* samples not present in the dataset, leading to inaccurate value function estimation for unseen state-action pairs and resulting in significant extrapolation error (Fujimoto et al., 2019). For example, in an autonomous driving application, the vehicle controlled by such a policy may fail to respond appropriately to a rare combination of road conditions, endangering the safety of passengers and other road users.

To address this issue, existing methods can be broadly classified into two categories: conservative estimation (Wu et al.,

2019; Kumar et al., 2020; Fujimoto & Gu, 2021; Kostrikov et al., 2021; Yu et al., 2021; Lyu et al., 2022) and uncertainty estimation (Yu et al., 2020b; Wu et al., 2021; An et al., 2021). Conservative estimation methods enforce consistency between the behavioral policy and the learned policy by adding a KL divergence constraint or a pessimistic penalty to the learned value function. This encourages the learned policy to maintain a pessimistic estimate for out-of-distribution state-action pairs and avoid them in online environments. However, conservative estimation methods tend to be overly conservative by avoiding unknown state-action pairs and their performance is severely restricted by the quality of the offline dataset. Uncertainty estimation methods measure the uncertainty of state-action pairs using an uncertainty metric and then correspondingly adjust the value function estimation. This balances the risk and reward of unseen state-action pairs. However, uncertainty estimation methods face risks of inaccurate uncertainty quantification and poor generalization when dealing with unseen state-action pairs, leading to potential negative consequences in safety-critical applications such as autonomous driving. Both conservative estimation and uncertainty estimation methods remain constrained by the closed and static offline dataset, and thus are incapable of effectively adapting to the environment.

To overcome this limitation, we propose a novel offline reinforcement learning paradigm, called Test-time Adapted Reinforcement Learning (TARL). This paradigm empowers offline RL to establish an interface with the environment, enabling the efficient update of a few parameters. Through the utilization of test-time data in the testing stage, it becomes possible to further fine-tune the model. As a result, the performance of offline RL can be significantly enhanced, and the learned policy can be effectively optimized to adapt to the real testing environment. Note that since we do not conduct an exploration of the environment and do not require feedback from the environment, this remains an offline rather than an online approach. TARL still retains the advantages of offline reinforcement learning in terms of efficiency and security. Specifically, we construct unsupervised test-time optimization objectives for discrete control and continuous control tasks separately. We fine-tune the parameters in the normalization layers of the policy by minimizing the output entropy or output uncertainty of the policy. In this manner, we can make minor adjustments to the policy using only the unlabeled data from the test environment, enabling it to adapt to the online testing environment. Moreover, we propose a debiasing term by KL divergence, which restricts the model parameters from becoming excessively large. This can effectively prevent the policy from overly focusing on the test-time data and thus avoid the bias problem caused by overfitting and error accumulation. In this paper, our main contributions are as follows:

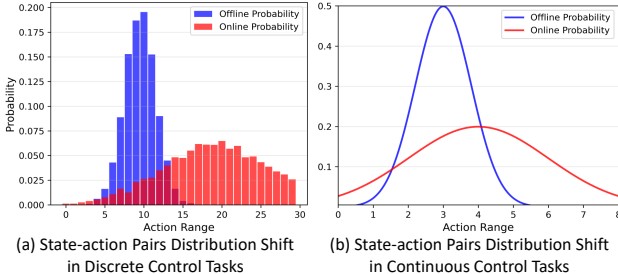

(a) State-action Pairs Distribution Shift in Discrete Control Tasks
(b) State-action Pairs Distribution Shift in Continuous Control Tasks

*Figure 1.* State-action pairs distribution shift. The horizontal axis represents the action value, the vertical axis denotes the action probabilities. The red color represents the actions in the offline data, and the blue color represents the actions in the online environment. The offline dataset and the dynamic online environment have a significant data distribution mismatch, called the out-of-distribution (OOD) problem. This problem hinders the application of reinforcement learning algorithms in real-world scenarios due to suboptimal policy performance.

- We propose a novel Test-time Adapted Reinforcement Learning method, referred to as TARL. This simple yet effective paradigm enables offline reinforcement learning methods to adapt to the real-world environment, with only a few parameters in normalization layers requiring updates during testing. This method can be applied to various existing offline reinforcement learning methods. By integrating TARL, these existing methods can overcome the limitations imposed by the static offline datasets and better handle the uncertainties and variations in the real-world environment.

- TARL combines the adaptability of test-time fine-tuning with the conservatism of offline learning. We separately design unsupervised optimization objectives for discrete control tasks and continuous control tasks to achieve test-time adaptation. By using unsupervised objectives, TARL adapts the offline policy based solely on the test data, avoiding the need to wait for environmental rewards. Therefore, TARL preserves the stability and safety of conservative offline reinforcement learning.

- We conduct extensive experiments on the Atari and D4RL benchmarks to demonstrate the effectiveness of our proposed method. Our method achieves stable improvement over the baseline on various discrete and continuous control tasks. Notably, on the hopper-expert-v2 task, our method achieved a remarkable 13.6% relative increase in episode return compared to CQL. These results demonstrate the robustness and efficacy of our method in improving the performance of offline reinforcement learning on a variety of tasks.

## 2. Related Work

### 2.1. Offline Reinforcement Learning

Due to the distribution shift between the offline dataset and the online environment, offline reinforcement learning often encounters issues related to out-of-distribution (OOD) state-action pairs. Prior works have attempted to address this problem by adding KL-divergence (Jaques et al., 2019; Peng et al., 2019), Wasserstein distance (Wu et al., 2019), or MMD (Kumar et al., 2019) between the learned policy and the behavior policy to avoid the OOD state-action pairs. However, these methods often require a separately estimated model of the behavior policy and are limited by their ability to accurately estimate the unknown behavior policy, especially when data is collected from multiple sources (Levine et al., 2020). Conservative Q-learning (CQL) (Kumar et al., 2020) does not require estimating the behavior policy, which is sought after by researchers.

Previous approaches to offline RL have also attempted to estimate uncertainty to determine the trustworthiness of Q-value predictions (An et al., 2021; Lee et al., 2022), but these methods have not been generally performant due to the high-fidelity requirements of uncertainty estimates in offline RL. (An et al., 2021)used a diversified q-ensemble to capture Q-value variance for uncertainty, yet it led to overfitting with scarce or biased datasets, harming uncertainty reliability and decision-making in novel situations. (Lee et al., 2022)faced challenges in exploration-exploitation balance. Its pessimistic bias to handle unseen states-action pairs sometimes made the agent overly avoid exploration, stalling learning and preventing optimal performance. Our approach is devised with the aim of empowering offline reinforcement learning methods to smoothly adapt to out-of-distribution (OOD) states-action pairs in the actual environment, and notably, it manages to achieve this without having to fall back on online rewards.

### 2.2. Test-Time Adaptation

Test-time adaptation (TTA) aims to improve model accuracy on out-of-distribution (OOD) test data by adapting the model with test samples. Existing TTA methods, such as TTT (Wang et al., 2020) and TTT++ (Liu et al., 2021), rely on joint training of a source model using both supervised and self-supervised objectives, followed by adaptation with a self-supervised objective at test time. However, this pipeline assumes a specific manner of model training, which may not always be controllable in practice. To address this, fully test-time adaptation methods have been proposed, which adapt a model with only test data. These methods include batchnorm statistics adaptation (Nado et al., 2020), test-time entropy minimization (Wang et al., 2020; Niu et al., 2022), prediction consistency maximization over different augmentations (Zhang et al., 2022).

The test-time adaptation method can be used to solve various distribution deviation problems, including the distribution deviation between offline datasets and online environments in offline reinforcement learning. In this paper, we use the test-time adaptation method to address this problem and improve the performance of offline policies by incorporating online data. Our approach aims to overcome the distribution shift problem by adapting the offline policy with test-time adaptation using online data.

## 3. Test-Time Adapted Reinforcement Learning

**Notations.** We represent the environment as a Markov Decision Process (MDP) consisting of a 5-tuple $< \mathcal{S}, \mathcal{A}, \mathcal{P}, R, \gamma >$, where $\mathcal{S}$ is the state space, $\mathcal{A}$ is the action space, $P(s'|s, a)$ is the transition probability distribution, $R : \mathcal{S} \times \mathcal{A} \to \mathbb{R}$ is the reward function, and $\gamma \in [0, 1]$ is the discount factor. Reinforcement learning (RL) aims to seek a policy $\pi(a|s)$ from the set of policy functions $\pi$ to maximize the expected cumulative discounted reward.

**Problem definition.** In offline reinforcement learning, the offline dataset $\mathcal{D} = \{(s_i, a_i, r_i, s_{i+1})\}_{i=1}^{N}$ is collected by a behavior policy $u(\cdot|s)$. The agent can only learn the offline policy $\pi^{off}(a|s)$ from the offline dataset $\mathcal{D}$, without interacting with the environment to improve the policy. A major challenge arises from the pervasive presence of Out-of-Distribution (OOD) state-action pairs in real-world environments, where offline policies encounter novel interactions beyond their training distribution. Due to the distribution shift between the state-action distribution $d^{on}(s, a)$ of online samples and $d^{off}(s, a)$ of offline samples, the learned offline policy $\pi^{off}(a|s)$ cannot be well adapted to the online environment (Lee et al., 2022). This lack of adaptability can trigger a series of adverse effects, such as poor performance and erratic decisions in real-world environments.

**Overview of TARL.** To solve the problem, we develop a Test-time Adapted Reinforcement Learning (TARL) to mitigate the distribution shift between the offline dataset and the online environment, as shown in Figure 2. Firstly, TARL computes the probability entropy. We set a threshold for the entropy to exclude out-of-distribution (OOD) samples. Subsequently, we use the selected highly confident samples and minimize the entropy loss to achieve unsupervised test-time fine-tuning. This process allows our model to adapt to the test-time data without relying on additional supervision, leveraging the inherent information within the data itself. Secondly, we compute the debiasing term by enforcing the KL divergence between the original policy and the test-time updated policy. By constraining the update, we prevent the network from overfitting to the test-time data and ensure its generalization ability. Lastly, we combine the entropy minimization term and the debiasing term to update only the layer normalization layers. By selectively updating these

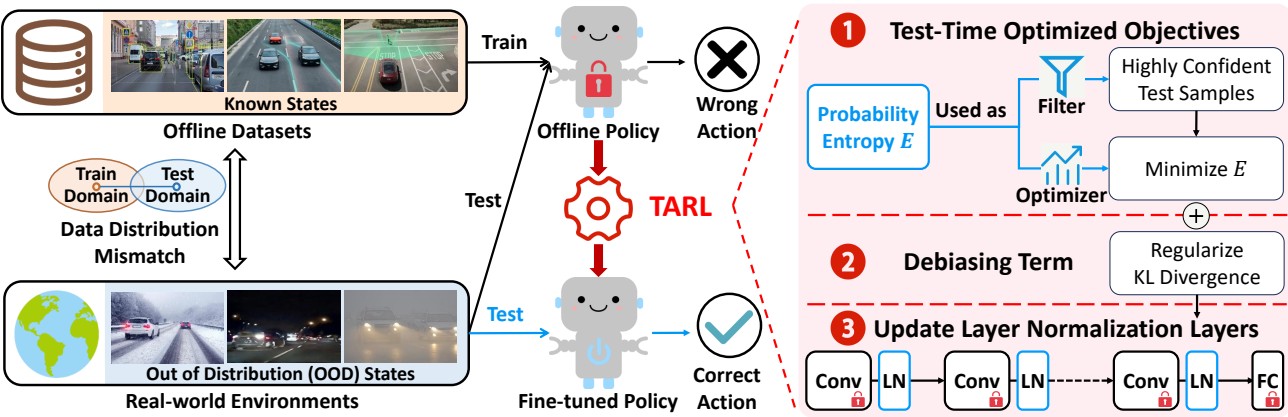

*Figure 2.* The overview of our Test-time Adapted Reinforcement Learning (TARL) method. TARL adjusts the trained policy during test time efficiently and effectively. Given a state from the test environment, our test-time adaptation strategy consists of three steps. **First**, we compute the probability entropy used as sample filters and test-time learning objectives. **Second**, we compute the debiasing term by enforcing the KL divergence between the trained policy and test-time updated policy. **Last**, we combine the entropy minimization term and the regularization term to update the layer normalization layers *only*.

layers, we can efficiently adapt the model to the test-time environment while preserving the knowledge learned during the pre-training phase. When the pre-trained offline policy is deployed in the online environment, we use the test data to adapt it, which enables the offline policy to better align with the distribution of the online environment, thereby leading to higher returns.

### 3.1. Significance of Offline RL Test-Time Adaptation

In reinforcement learning, a prominent out-of-distribution (OOD) problem often exists between the offline dataset and the online environment, as shown in Figure 1. The offline dataset is usually collected under specific, limited conditions, like past experiments with fixed parameters. Conversely, the online environment is dynamic and affected by unpredictable factors such as system state changes and evolving user behaviors. These differences cause a significant data distribution mismatch. As a result, deploying a pre-trained offline policy directly to the online environment may result in the agent taking unknown actions, leading to potential safety hazards due to the distributional shift between the online environment and the offline dataset. In safety-critical scenarios, such as autonomous driving, the agent taking unknown actions is not permissible. Therefore, it is risky to apply the original offline policy directly to the online environment. In contrast, online reinforcement learning enables the agent to interact directly with the environment, where the environmental rewards serve as crucial feedback mechanisms for policy improvement. However, in some online environments, acquiring rewards can be time-consuming or even infeasible. Real-world scenarios may not provide real-time reward signals to motivate the agent

to fine-tune the policy, further complicating the adaptation process. Therefore, it becomes crucial to enable the offline policy, which does not rely on real-time rewards from online environments, to adapt to real-world scenarios.

Motivated by this, we propose Test-time Adapted Reinforcement Learning (TARL). Inspired by online Test-Time Adaptation techniques, we construct unsupervised objective functions for discrete and continuous control tasks respectively. These functions are employed to fine-tune a minimal number parameters of layer normalization layers in the offline policy using test data. Specifically, our aim is to enable the policy to enhance the confidence of high-probability categories and avoid selecting low-confidence categories. Intuitively, the uncertainty associated with unknown out-of-distribution (OOD) actions, i.e., those actions not present in the offline dataset, is greater than that of known actions, which are the actions that exist within the offline dataset. Through the test-time fine-tuning, the offline policy can be encouraged to select actions with higher confidence. This, in turn, helps to avoid some of the unknown OOD state-action pairs, thereby improving the policy's performance and adaptability in the face of distributional shifts. By implementing this approach, we eliminate the reliance on reward signals for policy enhancement and empower the offline policy to adapt to the online environment. The proposed TARL approach has the potential to deftly circumvent the inherent limitations plaguing offline reinforcement learning. Through establishing interactions between the offline policy and the environment during test-time, our TARL promotes reinforcement learning to be more highly practical and efficacious in real-world scenarios.

### 3.2. Unsupervised Test-Time Learning Objectives

Our Test-time Adaptation Reinforcement Learning (TARL) is designed to establish interaction between offline reinforcement learning and online environments without relying on environmental reward signals. To achieve this goal, we propose unsupervised test-time learning objectives to update offline policies. The test-time objective $L_{ent}^t$ is to select the high confidence sample $s_t$ to update partial parameters of the policy. We understand that discrete continuous control tasks and continuous control tasks have distinct characteristics. Discrete control tasks involve choosing from a finite set of actions, like selecting a specific gear in a vehicle. Continuous control tasks, on the other hand, involve actions that can take on any value within a certain range, such as adjusting the speed of a robot. Therefore, we have tailored the test-time learning objectives for these two types of tasks respectively.

**Discrete Control Tasks.** For discrete control tasks, we use probability entropy to quantify the uncertainty of the actions predicted by the offline policy (Niu et al., 2022). As shown in the following formula:

$$E_{dis}(s; \Theta) = -\sum_{a \in \mathcal{A}} \pi_\Theta(a \mid s) \log \pi_\Theta(a \mid s), \quad (1)$$

where $\mathcal{A}$ is the action space, $s$ is the state in the online environment. $E_{dis}(s)$ is the probability entropy of the actions predicted by the policy network, which represents the uncertainty of taking an action.

A higher value of the entropy $E_{dis}(s)$ indicates a greater level of uncertainty in the actions predicted by the offline policy. As this uncertainty increases, there is a higher probability that the policy will execute unknown or incorrect actions, i.e., out-of-distribution (OOD) actions. In order to prevent the policy network from choosing those OOD actions with greater uncertainty, we filter out those samples with greater uncertainty by setting a threshold $E_0$:

$$f_{dis}(s) = \mathbb{I}_{\{E_{dis}(s;\Theta) < E_0\}}, \quad (2)$$

where $\mathbb{I}$ is the indicator function.

Then, we use the selected samples to update the policy network. To improve the offline policy's adaptability to the online environment for discrete control tasks, we minimize the $E_{dis}(s)$ as the unsupervised optimization objective. By reducing the uncertainty of the predicted actions, we can enhance the confidence of the offline policy in predicting correct actions during the test-time. Consequently, this enhances the policy's prediction accuracy and helps it better fit the online environment and handle new situations. What's more, for effective adaptation, we only update a few trainable parameters $\tilde{\Theta} \subseteq \Theta$ of all layer normalization layers. Thus the test-time optimized goal for the discrete control

task is as follows:

$$\mathcal{L}_{ent}^{dis} = \min_{\tilde{\Theta}} f_{dis}(s) E_{dis}(s; \Theta). \quad (3)$$

**Continuous Control Tasks.** Continuous control tasks are different from discrete control tasks. The output of the policy network is not about the probability of actions, thus we cannot directly use the minimum action probability entropy as the optimization goal. In continuous control tasks, the policy network outputs the specific value of each action. Across the entire action space, these outputs form a normal distribution, characterized by a mean $u$ and a variance $\sigma$. The action is then sampled by parameterization. So we adopt the following formula to represent the uncertainty:

$$E_{con}(s; \Theta) = \frac{1}{2} \left( \ln \left( 2\pi\sigma^2 \right) + 1 \right), \quad (4)$$

where $\sigma$ is the variance of the policy network, which represents the uncertainty of the action mean $u$.

Due to the distribution shift between the offline dataset and the online environment, the policy network will inevitably output OOD actions for a certain state. In order to filter out this part of OOD samples, we use a replay buffer to temporarily store the test data $s_i$. When a certain amount of data is collected, we use Formula (4) to calculate the uncertainty of all samples in the replay buffer $\mathcal{B} = \{s_i\}_{i=1}^N$ and select top k small variance sample data to fine-tune the strategy network. As shown in the following formula:

$$f_{con}(s) = \mathbb{I}_{\{E_{con}(s) \leq E_0\}}, \quad (5)$$

where $E_0$ is the k-th smallest entropy of $E_{con}(s_i)$.

By minimizing the above Formula (4), the policy network can output a more confident action mean $u'$ with less uncertainty. Thus the test-time optimized objective for the continuous control task is as follows:

$$\mathcal{L}_{ent}^{con} = \min_{\tilde{\Theta}} f_{con}(s) E_{con}(s; \Theta). \quad (6)$$

### 3.3. Test-Time Debiasing Regularization

Since the test-time finetuning employs unsupervised objectives, the generation of errors is inevitable. The network relies on the inherent characteristics within the test data for optimization. When the threshold $E_0$ is set higher, more online data are used to update the policy network. The more test data are involved, the more likely it is to introduce issues such as data noise and data distribution shifts, exacerbating the generation of errors. As the training progresses continuously and iteratively, these errors exhibit a cumulative effect. The accumulation of errors leads to a further negative impact on the model's performance. Moreover, the policy

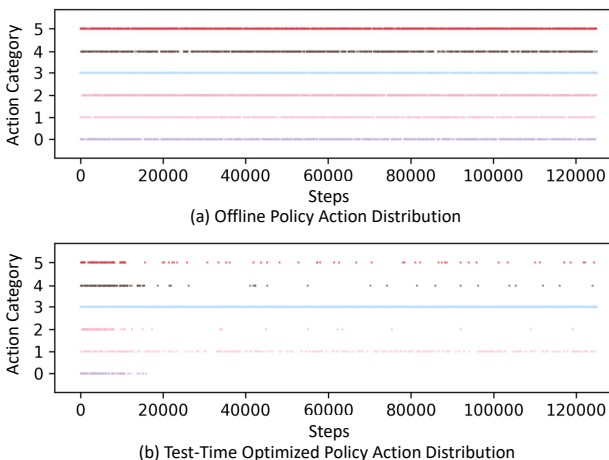

*Figure 3.* Policy network bias after test-time adaptation. When only using test-time learning objectives to update the offline policy, the policy network will tend to output a specific action category, such as the third category in the figure.

network tends to over-emphasize the local features within the test data while neglecting the overall data distribution. As a result, the policy network is prone to the problem of overfitting. The accumulation of errors and the overfitting to the test data are intertwined issues. Together, they give rise to an increasingly severe bias problem. As shown in Figure 3, the model may output with a high probability to a specific category. This bias can lead to suboptimal decision-making and a failure to generalize well to the real-world environment. In order to solve this problem, we propose to use the original pre-trained offline policy network $\pi^{off}$ whose model parameters are frozen and perform KL divergence constraints between the output of the fine-tuned offline policy $\pi^{tta}$ and the frozen policy network $\pi^{off}$. The KL Divergence loss is as follows:

$$\mathcal{L}_{kl}^t = KL(\pi^{off}||\pi^{tta}). \tag{7}$$

This debiased term by KL divergence effectively restricts the update of model parameters. As a result, the bias within the policy network is significantly reduced. With less bias and overfitting issues, the policy network can better generalize to different scenarios. During the test-time adaptation process, especially when dealing with out-of-distribution state-action pairs, the generalization ability of the policy is enhanced. It can make more accurate and reliable decisions, adapting well to various states and actions that may not be fully represented in the training data. This ultimately improves the overall performance and adaptability of the offline policy in real-world applications.

### 3.4. Debiased Test-Time Adaptation for RL

We use the state data $s_t$ during the interaction with the online environment to further fine-tune the policy network

---

**Algorithm 1** Training Method for TARL

---

**input** The trained offline policy $\pi^{off}$, the online state $s$, the replay buffer $\mathcal{B}$.
**output** The test-time updated policy $\pi^{tta}$.
1: Initialize the replay buffer $\mathcal{B}$ to temporarily store the state.
2: **for** episode = 1, M **do**
3:     Get the initialized state $s_0$ from the online environment.
4:     **for** step t = 1, T **do**
5:         Agent executes action $a_t$ and observes state $s_t$.
6:         Store state $s_t$ in $\mathcal{B}$.
7:         Filter out those states with greater uncertainty via Eqn. (2) or Eqn. (5).
8:         Compute the KL divergence between $\pi^{off}$ and $\pi^{tta}$ by Eqn. (8).
9:         Use filtered samples to update the trainable parameters $\hat{\Theta}$ in $\pi^{tta}$ via Eqn. (9).
10:     **end for**
11: **end for**

---

parameters so that the policy network can adapt to the online environment. The optimization objectives during iteration time step $t$ are as follows:

$$\mathcal{L}_{ent}^t = \begin{cases} \mathcal{L}_{ent}^{dis} & \text{if } \mathcal{A} \text{ is discrete action space,} \\ L_{ent}^{con} & \text{if } \mathcal{A} \text{ is continuous action space.} \end{cases} \tag{8}$$

Furthermore, we take into account the bias problem introduced during the test-time finetuning process. To address this issue, we proposed using the KL divergence loss as a debiasing term to regularize the policy network. We present the whole loss function used in the debiased test-time adapted reinforcement learning:

$$\mathcal{L}^t = \mathcal{L}_{ent}^t + \lambda * \mathcal{L}_{kl}^t, \tag{9}$$

where $\lambda$ is a hyperparameter that controls the degree of change of the output of the offline policy. When the $\lambda$ is large, the offline policy tends to the network that has not been updated; when the $\lambda$ is small, the offline policy parameter update step is almost determined by the online state $s_t$.

The overall method of our TARL is summarized in Algorithm 1. Leveraging the proposed debiased test-time learning objectives, TARL can update an offline reinforcement learning network by using test data without reward. During the test phase, TARL can analyze the test data to adjust its internal parameters, optimizing the decision-making strategies. TARL allows existing offline policies to effectively adapt to Out-of-Distribution (OOD) state-action pairs in the environment. Consequently, offline policies integrated with TARL can maintain high performance and stability in the face of unexpected environmental changes, significantly enhancing its generalization ability and practicality.

*Table 1.* Average episode return comparison of TARL against baseline methods on Atari benchmarks over the 10 evaluations. The highest mean scores are highlighted in bold.

| Algorithm | Qbert | Seaquest | Asterix | Pong | Breakout |
|---|---|---|---|---|---|
| REM | $914.38 \pm 6.81$ | $335.42 \pm 9.72$ | $387.00 \pm 4.68$ | $-20.23 \pm 0.10$ | $3.27 \pm 0.44$ |
| REM (TARL) | $\mathbf{920.18 \pm 15.82}$ | $\mathbf{339.26 \pm 9.38}$ | $\mathbf{394.73 \pm 8.53}$ | $\mathbf{-20.20 \pm 0.06}$ | $\mathbf{3.58 \pm 0.55}$ |
| QR-DQN | $646.35 \pm 17.08$ | $413.09 \pm 13.16$ | $503.38 \pm 14.60$ | $-18.48 \pm 0.12$ | $3.89 \pm 0.42$ |
| QR-DQN (TARL) | $\mathbf{672.73 \pm 14.44}$ | $\mathbf{424.57 \pm 11.91}$ | $\mathbf{819.24 \pm 30.74}$ | $\mathbf{-18.40 \pm 0.18}$ | $\mathbf{4.18 \pm 0.60}$ |
| CQL | $4334.64 \pm 259.67$ | $257.80 \pm 9.25$ | $504.59 \pm 13.98$ | $1.02 \pm 1.22$ | $8.26 \pm 1.17$ |
| CQL (TARL) | $\mathbf{4602.67 \pm 264.11}$ | $\mathbf{270.40 \pm 8.42}$ | $\mathbf{524.76 \pm 13.98}$ | $\mathbf{2.08 \pm 0.92}$ | $\mathbf{8.53 \pm 1.15}$ |

## 4. Experiments

To evaluate the effectiveness of Test-Time Adapted Reinforcement Learning (TARL), we conduct experiments on both discrete control and continuous control tasks.

### 4.1. Datasets

- **Atari Benchmark.** For discrete control tasks, we conduct experiments on Atari games (Bellemare et al., 2013). We evaluate our TARL on five Atari tasks: Qbert, Seaquest, Asterix, Pong, Breakout. These games provide diverse challenges through distinct mechanics, such as spatial reasoning in Qbert and reactive control in Pong.

- **D4RL Benchmark.** For continuous control tasks, we conduct experiments on D4RL benchmark (Fu et al., 2020). We evaluate our algorithm in five settings: Expert (optimal/near-optimal high-performing policy data, Fully Replay (training exclusively on fixed pre-collected datasets), Medium Policy (suboptimal policy-generated data), Medium Replay Buffer (mixed-quality data from training a medium policy), and Medium-Expert (hybrid expert-novice demonstrations). We evaluate TARL on three locomotion tasks: bipedal locomotion (HalfCheetah), monopedal jumping stability (Hopper), and dynamic balance maintenance (Walker2D).

### 4.2. Baselines

For discrete control tasks, following (Kumar et al., 2020), we compare TARL with three classic algorithms: QR-DQN (Dabney et al., 2018), REM (Agarwal et al., 2020) and CQL (Kumar et al., 2020). For continuous control tasks, we compare TARL with CQL (Kumar et al., 2020) and IQL (Kostrikov et al., 2022).

### 4.3. Evaluation Metrics

For the atari game, when deploying offline policy to the on-line environment, the agent interacts with the environment for 125,000 steps, each episode is not more than 27,000 steps, and the average episode return is used as the evaluation metric. For D4RL tasks, the agent interacts with the environment with 10 episodes, and we also use the average episode return as the metric. However, because the reinforcement learning prediction is very unstable, we repeat the above process 10 times and use the mean and variance of 10 times as the evaluation metric.

### 4.4. Implementation Details

We implement the discrete control experiments in atari following CQL (Kumar et al., 2020) and the continuous control tasks in the OfflineRL-Kit codebase (Sun, 2023)[1]. We use two distinct sets of hyperparameters for discrete control and continuous control tasks, respectively. All environments of the same type of task share the same hyperparameters. For the D4RL benchmark dataset with continuous control tasks, the hyperparameters used for all tasks were a learning rate of $1e^{-6}$, a buffer capacity size of 1000, and a selection of the top 10 small entropy samples to update the offline policy. The KL Divergence constraint $\lambda$ was set to 1.0. For the Atari dataset with discrete control tasks, we set the hyperparameters as follows: a learning rate of $1e^{-9}$, an entropy threshold $E_0$ of 0.1, and a KL Divergence constraint limit $\lambda$ of 1.5.

### 4.5. Offline RL on Discrete Control Tasks

Following (Kumar et al., 2020), we conducted experiments on discrete control tasks to compare the performance of our method with the classic offline reinforcement learning algorithms QR-DQN and CQL. As shown in Table 6, our method achieved a greater performance improvement than QR-DQN, and also outperformed the CQL method. In the Qbert task, TARL-enhanced REM (REM (TARL)) achieved an average episode return of 920.18, while the original REM had a return of 914.38. For QR-DQN, the average episode return increased from 646.35 to 672.73 when combined with TARL (QR-DQN (TARL)). In the Asterix task, the average episode return of CQL was 504.59, while CQL (TARL)

---

[1]The source code for this project is publicly available at https://github.com/xushoukai/TARL.

Table 2. Average episode return comparison of TARL against baseline methods on D4RL benchmarks. The highest mean scores are highlighted in bold.

| CQL Comparisons | | |
|---|---|---|
| Task Name | CQL | CQL (TARL) |
| hopper-expert-v2 | 99.72 | **113.34** |
| walker2d-fully-replay-v2 | 95.31 | **97.99** |
| walker2d-expert-v2 | 113.25 | **113.57** |

| IQL Comparisons | | |
|---|---|---|
| Task Name | IQL | IQL (TARL) |
| Walker2d-medium-v2 | 79.92 | **82.43** |
| Walker2d-expert-v2 | 110.31 | **110.49** |

Table 3. The effect of Entropy Threshold $E_0$. The highest mean scores are highlighted in bold.

| Entropy Threshold | CQL | TARL (Ours) |
|---|---|---|
| 0.9 | $1.02 \pm 1.22$ | $-19.98 \pm 0.11$ |
| 0.7 | $1.02 \pm 1.22$ | $-19.61 \pm 0.22$ |
| 0.5 | $1.02 \pm 1.22$ | $-18.79 \pm 0.42$ |
| 0.3 | $1.02 \pm 1.22$ | $-13.54 \pm 1.22$ |
| 0.1 | $1.02 \pm 1.22$ | $\mathbf{1.70 \pm 0.82}$ |

Table 4. Effectiveness of low-entropy selective training.

| Training Strategy | TARL |
|---|---|
| Global Entropy Minimization | 77.74 |
| Low-entropy Selective Training | **82.95** |
| High-entropy Selective Training | 72.91 |

reached 524.76. These results demonstrate the effectiveness of our method for discrete control tasks. Using test-time data and constructing unsupervised test-time optimization objectives, TARL fine-tunes the layer normalization layers of the offline policies. This allows the existing offline RL algorithms to better adapt to the distribution of the online environment. This further validates that our TALR can be applied to various existing offline reinforcement learning methods to make them adapt to online environments.

### 4.6. Offline RL on Continuous Control Tasks

Our experimental results in Table 2 demonstrate consistent performance improvements over both CQL and IQL baselines across D4RL benchmarks. The method achieves substantial gains of 2.68 and 13.62 on CQL's walker2d-fully-replay-v2 and hopper-expert-v2 tasks respectively, while maintaining stable superiority (0.32) even on expert-level walker2d-expert-v2, suggesting enhanced capability to handle policy mismatch across varying dataset qualities. Similar improvements emerge with IQL baselines, where our approach outperforms by 2.51 on Walker2d-medium-v2, while preserving a marginal but consistent advantage (0.18) on Walker2d-expert-v2. This performance pattern reveals two critical insights: first, the method demonstrates stronger efficacy when tackling medium-quality datasets with pronounced distribution shifts. Second, the persistent albeit smaller improvements on expert-level tasks confirm the universal existence of offline-online distribution discrepancies regardless of behavior policy quality. These findings collectively validate our approach's robustness in mitigating distribution shift challenges, particularly in suboptimal data regimes where conventional offline RL methods struggle. Extended experimental results are provided in the supplementary materials A, including comprehensive comparisons across more tasks.

### 4.7. Ablation Study

**Effect of Entropy Threshold** $E_0$**.** We verify the effect of entropy threshold $E_0$ on test-time adapted RL under the pong game in the Atari benchmark. The learning rate is $1e^{-9}$. From Table 3, we can clearly see that when the entropy threshold increases from 0.3 to 0.5, the performance of test-time adapted RL declines, which just shows that just minimizing the test-time optimized goal does not necessarily promote the offline policy to adapt to the online environment, it is necessary to have a suitable threshold to filter some samples with relatively high confidence to update the offline policy to adapted the online environment.

**Effect of Selective Training on Low-entropy States.** The experiments were conducted in the walker2d-medium-v2 task to evaluate the hypothesis that selective training on low-entropy states improves TTA performance. The experimental setup included three conditions:

- Global Entropy Minimization, where all available data were used for test-time adaptation.
- Low-entropy Selective Training, where only samples with entropy below a predefined threshold were used for training.
- High-entropy Selective Training, where only samples with entropy over a predefined threshold were used for training.

Our results demonstrate that selective training on low-entropy states improves TTA performance more effectively than global entropy minimization. Meanwhile, if we only select high-entropy samples for tta, the performance actually becomes worse. This further indicates that selective training on low-entropy states enables beneficial knowledge transfer.

**Effect of KL divergence** $\lambda$**.** To investigate the impact of KL divergence on offline policy constraints, we conducted experiments on the Pong game with a learning rate of $1e^{-9}$ and an entropy threshold of 0.5. Table 5 shows that when the

*Table 5.* The effect of KL divergeence $\lambda$. The highest mean scores are highlighted in bold.

| $\lambda$ | CQL | CQL (TARL) |
|---|---|---|
| 0.5 | $1.02 \pm 1.22$ | $0.92 \pm 1.08$ |
| 1.0 | $1.02 \pm 1.22$ | $1.32 \pm 1.39$ |
| 1.5 | $1.02 \pm 1.22$ | $\mathbf{2.08 \pm 0.92}$ |
| 2.0 | $1.02 \pm 1.22$ | $0.62 \pm 1.73$ |

entropy threshold is set too high, more uncertain samples $s_t$ are used to update the offline policy, which can make it difficult for the policy to adapt to the online environment. Incorporating an appropriate KL divergence constraint between the offline policy and the frozen base policy can help the offline policy transition smoothly to the online environment by setting a more suitable $\lambda$ in the optimization goal during the adaptation process. When $\lambda$ changes from 0.5 to 1.0, we can see that the adaptive performance is improving, but when $\lambda$ is too large, the adaptive performance will deteriorate.

## 5. Conclusion

In this work, we study how to use the test-time data to learn a promising policy in the offline reinforcement learning task. To this end, we propose Test-time Adapted Reinforcement Learning (TARL) to efficiently and effectively adjust the parameters of the layer normalization layers. We first introduce an entropy minimization loss as the unsupervised training objective for policy parameter update. Then we propose a debiasing term that regularizes the KL divergence of predictions between the original pre-trained policy and the test-time updated policy to avoid model degeneration due to the overestimation of a few test-time inputs. Extensive experiments on the widely-used Atari game benchmarks show the superior performance of our method.

## Acknowledgments

This work was partially supported by the Joint Funds of the National Natural Science Foundation of China (Grant No.U24A20327), Key-Area Research and Development Program Guangdong Province 2018B010107001, the Major Key Project of Peng Cheng Laboratory (PCL) PCL2023A08, and TCL Science and Technology Innovation Fund, China.

## Impact Statement

The proposed TARL enables offline reinforcement learning to interact with online environments using only test data. TARL allows existing offline policies to effectively adapt to Out-of-Distribution (OOD) state-action pairs in the environment. Consequently, offline policies integrated with TARL can maintain high performance and stability in the face of unexpected environmental changes, significantly enhancing its generalization ability and practicality.

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

# SUPPLEMENTARY MATERIALS

## A. More Comparisons on D4RL with CQL

We compared the performance of our method with the classic offline RL algorithm CQL on more tasks. As shown in Table 6, our method outperformed CQL significantly. Moreover, we observed that our method achieved greater performance improvement than the baseline when the offline dataset was collected using a medium strategy. For instance, on the walker2d-m task, our method achieved a return that was 5.51 higher than CQL. This demonstrates that our method has a significant advantage over normal offline reinforcement learning algorithms when there is a large distribution shift between the offline dataset and the online environment. On the dataset collected using the expert strategy, our method performed similarly to CQL, but still showed some improvements. This indicates that regardless of how the behavior strategy collects data, there is always a certain distribution offset between the offline dataset and the online environment.

*Table 6.* Normalized average score comparison of TARL against baseline methods on D4RL benchmarks over the 10 evaluations. The score ranges from 0 (random policy) to 100 (expert policy). The abbreviations used are m for medium-v2, m-r for medium-replay-v2, and m-e for medium-expert-v2. The highest mean scores are highlighted in bold.

| Task Name | CQL | CQL (TARL) |
|---|---|---|
| halfcheetah-m | $50.67 \pm 0.30$ | $\mathbf{50.75 \pm 0.11}$ |
| hopper-m | $85.74 \pm 4.42$ | $\mathbf{87.79 \pm 3.96}$ |
| walker2d-m | $77.44 \pm 5.50$ | $\mathbf{82.95 \pm 3.00}$ |
| halfcheetah-m-r | $46.82 \pm 0.26$ | $\mathbf{46.92 \pm 0.23}$ |
| hopper-m-r | $101.47 \pm 0.11$ | $\mathbf{101.55 \pm 0.16}$ |
| walker2d-m-r | $85.13 \pm 6.54$ | $\mathbf{85.61 \pm 4.33}$ |
| halfcheetah-m-e | $94.36 \pm 2.57$ | $\mathbf{95.12 \pm 0.39}$ |
| hopper-m-e | $106.18 \pm 4.27$ | $\mathbf{107.63 \pm 3.29}$ |
| walker2d-m-e | $112.73 \pm 1.98$ | $\mathbf{113.64 \pm 0.20}$ |
| Avg. Score | 84.52 | **85.79** |

## B. More Discussion about the Difference between the TARL and Online RL

TARL and online reinforcement learning (Xie et al., 2021; Lee et al., 2022) operate at different levels. The fundamental distinction between TARL and online RL lies in their operational paradigms and feedback dependencies.

When offline RL methods run in online environments, they will suffer from OOD issues. The core motivation of TTA is that environmental dynamics and distribution shifts cause performance drops in offline-trained models. TARL aims to maintain stability amidst data distribution changes. TARL fine-tunes the policy during the test-time phase through entropy minimization, without needing environment feedback. It effectively adapts to changes in test data distribution and ensures efficient updates. In contrast, online RL algorithms depend on environment feedback for policy updates. However, in some environments acquiring rewards can be time-consuming or even infeasible. Therefore, offline-to-online RL cannot update policies.

