# OpenReview forum: "Test-time Adapted Reinforcement Learning with Action Entropy Regularization"
_ICML.cc/2025/Conference — ICML 2025 poster_

### Official Review · Reviewer_k3Mg · 2025-03-02

**Overall Recommendation:** 2

**Summary:**

This submission proposes Test-time Adapted Reinforcement Learning (TARL) to address the transfer gap between offline learning and online testing. TARL has two main components: minimizing the entropy of action probabilities by filtering actions with high entropy and efficiently updating only the layer normalization parameters, along with a KL divergence constraint between the fine-tuned policy and the original policy. The authors empirically demonstrate the effectiveness of the proposed TARL.

**Claims And Evidence:**

The claims made in the submission are supported by clear and convincing evidence.

**Essential References Not Discussed:**

This submission includes sufficient related references.

**Experimental Designs Or Analyses:**

The experimental design and analysis exhibit soundness and validity.

**Methods And Evaluation Criteria:**

The proposed methods and the corresponding evaluation criteria are appropriate for addressing the problem.

**Other Comments Or Suggestions:**

No further comments and suggestions. Please see the following questions.

**Other Strengths And Weaknesses:**

**Strengths**

The main issue discussed in this paper, the transfer gap between offline learning and online testing, is crucial in the field of reinforcement learning. The proposed TARL method is articulated clearly, offering a well-structured approach.

**Weaknesses**

The improvements reported on most Atari and D4RL tasks are relatively marginal. This raises questions about the practical significance and robustness of the proposed method under different conditions and task complexities.

**Questions For Authors:**

1. The motivation for only fine-tuning the layer normalization parameters is not entirely clear. Why was this particular choice made? Could you provide more insight into the rationale behind this decision?
2. In the comparison results of Table 1 and Table 2, are the baselines also fine-tuned with test data?
3. This work addresses the issue of the transfer gap between offline learning and online environments. There are several offline-to-online algorithms[1][2] that seem very related to this work. Shouldn't these algorithms be compared within the study?
4. While CQL is a classic offline RL algorithm, it would be valuable to apply the proposed method to more powerful algorithms, such as DT[3] and IQL[4].
5. I am curious about the sensitivity of the proposed TARL to the hyperparameters. The detailed hyperparameter settings for different tasks should be included for clarity.

[1] Xie T, Jiang N, Wang H, et al. Policy finetuning: Bridging sample-efficient offline and online reinforcement learning[J]. Advances in neural information processing systems, 2021, 34: 27395-27407.
[2] Lee S, Seo Y, Lee K, et al. Offline-to-online reinforcement learning via balanced replay and pessimistic q-ensemble[C]//Conference on Robot Learning. PMLR, 2022: 1702-1712.
[3] Chen L, Lu K, Rajeswaran A, et al. Decision transformer: reinforcement learning via sequence modeling, 2 June 2021[J]. URL http://arxiv. org/abs/2106.01345, 2021.
[4] Kostrikov I, Nair A, Levine S. Offline reinforcement learning with implicit q-learning[J]. arXiv preprint arXiv:2110.06169, 2021.

**Relation To Broader Scientific Literature:**

This work primarily focuses on the field of offline reinforcement learning.

**Theoretical Claims:**

This submission does not provide theoretical claims.

---

> ### Author Rebuttal · Authors · 2025-04-01
>
> >Q1. The improvements reported on most tasks are relatively marginal. This raises questions about the practical significance and robustness under different conditions and task complexities.
>
> A1. Thanks for your valuable feedback. The marginal improvements should be interpreted through four aspects.
>
> 1. Our performance reflects a trade-off between efficiency and performance. TARL updates only LN parameters during testing, requiring no gradient updates to the full network.
>
> 2. Unlike online RL, TARL operates purely on test samples without reward signals, making it suited for safety-critical applications where exploration is prohibited. TARL preserves the stability and safety of conservative offline RL.
>
> 3. TARL shows improvements across discrete (Atari) and continuous (D4RL) control tasks, which indicates its robustness.
>
> 4. We study more extensively across multiple datasets to validate our effectiveness.
> From Table D, TARL achieves significant improvement.
>
> Table D. Comparison of TARL against the baseline.
> Task|CQL|CQL(TARL)|
> |-|-|-|
> |Walker2d-fully-replay-v2|83.38|**97.99**|
> |Hopper-expert-v2|99.72|**113.34**|
> |Antmaze-medium-play-v0|6|**7**|
> |Antmaze-medium-diverse-v0|2|**5**|
>
> >Q2. The motivation for only fine-tuning the layer normalization parameters is not entirely clear. Why was this particular choice made?
>
> A2. The test-time phase requires efficient model parameter updates to ensure the model can adapt to the new environment promptly. Thus, TARL only updates parameters in layer normalization layers of the model, ensuring computational efficiency without full model gradient updating. This allows for rapid adaptation while maintaining model stability.
>
> >Q3. In Table 1 and 2, are the baselines also fine-tuned with test data?
>
> A3. The baselines do not fine-tuned with test data.
>
> >Q4. This work addresses the issue of the transfer gap between offline learning and online environments. There are several offline-to-online algorithms[1][2] that seem very related to this work. Shouldn't these algorithms be compared within the study?
>
> A4. Comparing TARL directly with online RL methods isn't appropriate as TARL is an offline RL method. TARL and offline-to-online RL operate at different levels.
>
> When offline RL methods run in online environments, they will suffer from OOD issues. The core motivation of TTA is that environmental dynamics and distribution shifts cause performance drops in offline-trained models. TARL aims to maintains stability amidst data distribution changes. TARL fine-tunes the policy during the test-time phase through entropy minimization, without needing environment feedback. It effectively adapts to changes in test data distribution and ensures efficient updates.
> In contrast, offline-to-online RL algorithms[1][2], depend on environment feedback for policy updates. However, in some environments acquiring rewards can be timeconsuming or even infeasible. Therefore, offline-to-online RL cannot update policies.
>
> We will include and discuss relevant literatures in the revised manuscript.
>
> [1] Policy finetuning: Bridging sample-efficient offline and online reinforcement learning. NeurIPS, 2021.
>
> [2] Offline-to-online reinforcement learning via balanced replay and pessimistic q-ensemble. PMLR, 2022.
>
> >Q5. While CQL is a classic offline RL algorithm, it would be valuable to apply the proposed method to more powerful algorithms, such as DT[3] and IQL[4].
>
> A5. Accoding to your suggestion, we present a comparative analysis of IQL[4] performance in Table E. Even on the expert, we still achieve improvement.
>
> Table E. Effectiveness of TARL applied to IQL.
> |Task|IQL|IQL(TARL)|
> |-|-|-|
> |Walker2d-medium-v2|79.92|**82.43**
> |Walker2d-expert-v2|110.31|**110.49**
>
> However, applying TARL to DT[3] would require higher workload due to the inherent differences in the Transformer architecture. Given the time constraints of this submission, we are unable to complete the experiments. We will explore this direction in future work.
>
> [3] Decision transformer: reinforcement learning via sequence modeling. NeurIPS 2021.
>
> [4] Offline reinforcement learning with implicit q-learning. ICLR 2022.
>
> >Q6. Curiou about the sensitivity of the proposed TARL to the hyperparameters. The detailed hyperparameter settings for different tasks should be included for clarity.
>
> A6. We use two distinct sets of hyperparameters for discrete control and continuous control tasks, respectively. All environments of the same type of task share the same hyperparameters.
> This further highlights the strong generalization capability of TARL. We have show hyperparameter settings in Section 4.4 and would make them more detailed and clearly.
>
> For important hyperparameters entropy threshold $E_0$ and the KL divergence $\lambda$, we have conducted ablation studies in Section 4.7. The TARL is sensitive to $E_0$ because a small $E_0$ is crucial for selecting high-confidence samples.
>
> ---
> We sincerely hope our clarifications above have addressed your questions.

---

> > ### Comment · Reviewer_k3Mg · 2025-04-04
> >
> > Thank you for your thoughtful response. I still have some follow-up questions:
> >
> > 1. Regarding the experimental settings for Tables 1 & 2: Did you first run the original baseline on the offline dataset and then apply the proposed TARL method to the baseline for conducting test-time adaptation?
> >
> > 2. Regarding the fine-tuning of the LN layer: I noticed that it is more efficient to update only the LN layer during test-time. What I would truly like the authors to discuss is the rationale behind fine-tuning the LN layer specifically, as opposed to other parameters in the model. Additionally, for the continuous tasks, could you elaborate on how the LN layer was integrated into the network?

---

> > > ### Author Response · Authors · 2025-04-07
> > >
> > > >Q1. Regarding the experimental settings for Tables 1 & 2: Did you first run the original baseline on the offline dataset and then apply the proposed TARL method to the baseline for conducting test-time adaptation?
> > >
> > > A1. Thanks for your valuable feedback. To directly address your question: Yes, we first run the original baseline on the offline dataset and then apply our TARL method to the baseline for conducting test-time adaptation.
> > >
> > > We begin with a pre-trained offline RL policy that has conventional offline training on the dataset. This policy serves as the initialization for deployment. During testing phase, we apply TARL's unsupervised optimization to *enhance the pre-trained policy* without modifying its original training process.
> > >
> > > This design ensures TARL functions as a universal test-time enhancement approach, improving pre-trained policies' robustness to distribution shifts while preserving their original training integrity. The performance gains in Tables 1-2 stem solely from this test-time adaptation, not from retraining baselines.
> > >
> > > >Q2. Regarding the fine-tuning of the LN layer: I noticed that it is more efficient to update only the LN layer during test-time. What I would truly like the authors to discuss is the rationale behind fine-tuning the LN layer specifically, as opposed to other parameters in the model.
> > >
> > > A2. The rationales behind fine-tuning the LN layer specifically are as follows:
> > >
> > > **1. LN's Unique Role in Distribution Calibration**
> > >
> > > LayerNorm layers (h' = γ ⊙ (h - μ)/σ + β) directly interact with input data statistics (mean, variance) and are inherently sensitive to distribution shifts. Tuning LN layers allows learnable affine transformations (γ,β) to absorb data distribution shifts. This makes LN parameters the optimal adaptation knobs for distribution mismatches.
> > >
> > > **2. Parameter Isolation in Normalization Layers**
> > >
> > > The parameters of LN layers are inherently localized and independent, unlike the tightly coupled parameters of other layers (e.g., convolutional or linear layers).
> > >
> > > **Other Layers (Strong Coupling):**
> > > Parameters in convolutional or fully connected layers are globally interdependent. Adjusting even a single layer’s weights can cascade through the network, disrupting hierarchical feature representations.
> > >
> > > **LN Layers (Weak Coupling):**
> > > LN’s scaling (γ) and shifting (β) parameters act as local calibrators for feature distributions. They normalize activations per sample and do not depend on batch statistics or cross-layer interactions. This design ensures:
> > >
> > >   • **Localized Impact:** Adjusting γ/β affects only the current layer’s output scale and offset, preserving the pre-trained model’s core feature extraction.
> > >
> > >   • **Statistical Independence:** LN’s per-sample normalization avoids batch-size dependencies, making it robust to dynamic test environments (e.g., single-sample batches or mixed distribution shifts).
> > >
> > > >Q3. For the continuous tasks, could you elaborate on how the LN layer was integrated into the network?
> > >
> > > A3. In our method, LN layers are integrated into the hidden layers of the actor and critic network backbones for both discrete and continuous tasks. For continuous control tasks, the final action outputs (e.g., mean and variance for Gaussian policies) are generated by a separate linear output layer. Notably, this final linear output layer remains devoid of LN integration.
> > >
> > > ---
> > > We sincerely hope our clarifications above have addressed your questions.

---

### Official Review · Reviewer_Hvb2 · 2025-03-09

**Overall Recommendation:** 5

**Summary:**

This paper presents Test-time Adapted Reinforcement Learning (TARL) for the distribution shift issue of offline RL. TARL creates unsupervised test-time objectives for different control tasks. Moreover, it uses a KL divergence constraint to avoid bias. TARL only updates layer normalization parameters during testing for efficiency. Experiments on Atari and D4RL datasets show its superiority over baselines.

**Claims And Evidence:**

The claims made in the submission are supported by clear and convincing evidence.

**Essential References Not Discussed:**

The authors clearly discussed key methodologies in the field of efficient attention.

**Experimental Designs Or Analyses:**

TARL shows significant advantages in experiments across discrete and continuous control tasks. The experimental results support the claim that TARL can be applied to various existing offline RL methods, effectively enhancing their performance in online environments.

The experimental designs and analyses presented in the paper are sound and robust, but I still have some questions:
1. This paper mentions that there is a distribution bias between offline training and online testing in the abstract. In the experiments, where is this bias specifically reflected?
2. The details of the Atari and D4RL benchmarks are not clear. For example, the environment and the agent setting are unknown. It would be better to add more descriptions about the two benchmarks.

**Methods And Evaluation Criteria:**

TARL enables offline RL to establish an interface with the environment during the testing stage. It uses test data to fine-tune the model without the need for environmental rewards, simply by using the inherent information within the test data itself.
The proposed methods and evaluation criteria make sense for the problem, but I still have some questions:

1.	In Line 70, the authors claim that they propose a novel offline reinforcement learning paradigm. However, the method belongs to a test-time adaptation method. Is there a conflict between offline reinforcement learning and test-time adaptation? I suggest further clarification on this.
2.	Continuous control tasks assume that the action obeys a normal distribution. However, the actions in real applications may not be strictly normal. Can the TARL method based on the normal distribution assumption still work?

**Other Comments Or Suggestions:**

In Line 359, the period is missing.

**Other Strengths And Weaknesses:**

**Strengths**

1.	During test time, TARL updates only a few parameters of the LN layers. Compared to updating the entire network, adjusting only the LN layer parameters requires far fewer computations. This speeds up adaptation and enhances overall efficiency.
2.	TARL uses the KL divergence constraint as a debiasing term. This effectively restricts the update of model parameters, reducing the bias within the policy network and preventing overfitting to the test-time data.

**Weakness**

The performance of TARL depends on selecting appropriate entropy thresholds ($E_0$) and KL divergence weight ($\lambda$).

**Questions For Authors:**

N/A

**Relation To Broader Scientific Literature:**

TARL uses test-time data to fine-tune the model without environmental rewards. It helps the policy adapt to real-world test environments.

TARL can be applied to existing offline reinforcement learning methods. By integrating TARL, these methods can overcome the limitations imposed by static offline datasets. It helps them better handle the uncertain environment. This indicates a broad application prospect for the proposed method.

**Theoretical Claims:**

N/A

---

> ### Author Rebuttal · Authors · 2025-04-01
>
> >Q1. In Line 70, the authors claim that they propose a novel offline reinforcement learning paradigm. However, the method belongs to a test-time adaptation method. Is there a conflict between offline reinforcement learning and test-time adaptation? I suggest further clarification on this.
>
> A1. Thank you for your valuable feedback. Offline reinforcement learning and test-time adaptation are not conflicting concepts.
> TARL is a framework designed for test-time adaptation of offline reinforcement learning methods. Offline RL learns from pre-collected data without interacting with the real-time environment. TARL fine-tunes the offline-trained policy model using test data to better fit real-world conditions. It is important to note that TARL requires no environment interaction or feedback during adaptation. As a result, Offline RL algorithms optimized by TARL with test-time adaptation retain the offline characteristic, maintaining its efficiency and safety while improving real-world performance.
>
>
>
> >Q2. Continuous control tasks assume that the action obeys a normal distribution. However, the actions in real applications may not be strictly normal. Can the TARL method based on the normal distribution assumption still work?
>
> A2. In practice, TARL remains effective even when the action distribution is not a strictly normal distribution. The core principle of TARL is to reduce policy uncertainty during action selection, thereby improving stability and adaptability. This process does not depend on any specific state-action distribution. For non-normal distributions, entropy minimization still encourages the policy to favor more confident actions, leading to better performance.
>
>
> >Q3. This paper mentions that there is a distribution bias between offline training and online testing in the abstract. In the experiments, where is this bias specifically reflected?
>
> A3. In our experiments, the distributional shift primarily manifests as a mismatch between state-action distributions in offline training datasets and evaluation environments. This OOD stems from the sampling incompleteness problem: While the state-action space is theoretically infinite, offline datasets can only capture a finite subset through sampling. Consequently, the policy inevitably encounters unobserved state-action combinations during real-world deployment, highlighting the critical need for test-time adaptation mechanisms like TARL.
>
>
> >Q4. The details of the Atari and D4RL benchmarks are not clear. For example, the environment and the agent setting are unknown. It would be better to add more descriptions about the two benchmarks.
>
> A4. We provide additional details regarding the benchmarks used in our experiments.
>
> **Atari Benchmark:**  For **discrete control tasks**, we evaluate policy performance on five representative Atari environments: Qbert, Seaquest, Asterix, Pong, and Breakout. These games provide diverse challenges through distinct mechanics, such as spatial reasoning in Qbert and reactive control in Pong.
>
> **D4RL Benchmark:**  For **continuous control tasks**, we employ three standardized D4RL dataset configurations: Medium Policy (suboptimal policy-generated data), Medium Replay Buffer (mixed-quality trajectories), and Medium-Expert (hybrid expert-novice demonstrations). We evaluate TARL on three locomotion tasks: bipedal locomotion (HalfCheetah), monopedal jumping stability (Hopper), and dynamic balance maintenance (Walker2D).
>
> **Consistent Experimental Settings**
>
> Our implementation strictly follows original offline RL methodologies for fair comparison. For example, CQL(TARL) maintains identical network architectures and training hyperparameters to vanilla CQL.
>
>
> >Q5. The performance of TARL depends on selecting appropriate entropy thresholds ($E_0$) and KL divergence weight ($\lambda$).
>
> A5. The selection of appropriate entropy thresholds $E_0$ and KL divergence weights $\lambda$ is critical to the performance. This underscores the necessity of the data filtering mechanisms and debiasing regularization  introduced in our method. A suitable $E_0$ ensures that only high-confidence samples are retained, thereby preventing the introduction of noise from out-of-distribution data and preserving the stability and reliability of policy updates. Similarly, the selection of an optimal KL divergence weight $\lambda$ plays a key role in facilitating a smooth transition of the offline policy to the online environment.
>
> ---
> We sincerely hope our clarifications above have addressed your questions.

---

> > ### Comment · Reviewer_Hvb2 · 2025-04-09
> >
> > After carefully reviewing the authors' thorough rebuttal, I am convinced that they have effectively addressed my key questions and criticisms raised during the review process. Additionally, I also examine the other reviewers' comments. The expanded results rigorously validate the robustness and practical advantages of TARL across diverse benchmarks. Based on their comprehensive explanations and enhanced experimental validation, I have decided to raise my score.

---

### Official Review · Reviewer_xAD3 · 2025-03-10

**Overall Recommendation:** 3

**Summary:**

This paper introduces Test-Time Adapted Reinforcement Learning (TARL), a method designed to help offline RL policies adapt to distribution shift during deployment by leveraging test data—without needing additional reward signals. The core idea involves (1) learning objectives that minimize policy entropy for newly encountered states to reduce uncertainty (discrete tasks) or output variance (continuous tasks), (2) filtering out high-uncertainty (out-of-distribution) states so only lower-uncertainty states are used for adaptation, and (3) a KL divergence regularization term that keeps the adapted policy close to the original offline policy to prevent overfitting or degenerate solutions. Empirical results on Atari (discrete) and D4RL (continuous) benchmarks show that incorporating TARL on top of standard offline RL algorithms (e.g., CQL, QR-DQN, REM) improves final performance on out-of-distribution evaluation settings. The paper argues that TARL can be integrated with multiple offline RL methods by only updating the parameters in layer normalization layers, thus making test-time adaptation both computationally cheap and stable.

**Claims And Evidence:**

The core hypothesis underlying the TARL approach appears to be that selective training on low-entropy states facilitates information transfer to other similar states in a way that cannot be achieved through simpler entropy minimization strategies. However, this fundamental hypothesis is neither explicitly articulated nor well tested in the paper.

The proposed state-selective entropy minimization approach lacks critical comparisons to two obvious alternatives during runtime:
1. Simply decreasing the temperature globally in a Boltzmann policy
2. Using argmax action selection on states with low entropy

Without these comparisons, it's impossible to determine whether the reported improvements stem from the claimed state-selective mechanism or could be achieved through much simpler approaches that also reduce policy entropy. The paper presents performance improvements without convincingly demonstrating that these improvements are causally linked to the proposed state-selective training mechanism rather than to a simpler entropy reduction effect that could be achieved with less complex methods.

**Essential References Not Discussed:**

See "Claims And Evidence" section.

**Experimental Designs Or Analyses:**

As discussed in the "Claims And Evidence" and "Methods And Evaluation Criteria" sections, the experimental design cannot validate the central hypothesis because it lacks baselines for comparison. The experiments show that the method works in practice but fail to establish why it works or whether simpler alternatives might work equally well. The improvements shown could result from effective hyperparameter tuning of the entropy threshold rather than from the claimed state-selective mechanism. Refer to the "Methods And Evaluation Criteria" section for the specific comparisons and controlled experiments that would be needed to properly validate the paper's core hypothesis.

**Methods And Evaluation Criteria:**

While the paper's method of state-selective entropy minimization is coherent, the evaluation is fundamentally incomplete because it lacks the necessary experiments to validate the core hypothesis. To properly evaluate whether selective training on low-entropy states provides unique benefits, the following comparisons are essential:

1. Direct performance comparison with global temperature reduction in Boltzmann policies
2. Comparison with a policy that simply uses argmax action selection for states below an entropy threshold
3. State-specific analysis showing exactly where and how state-selective training performs differently than these simpler alternatives

The paper should provide:
- Controlled experiments in simplified environments where the hypothesized information transfer from low-entropy to related states can be explicitly tracked
- Concrete examples identifying specific state types or scenarios where state-selective training provides advantages that global approaches cannot
- Ablation studies comparing selective entropy minimization against global entropy minimization

Without these critical evaluations, it remains unclear whether the additional complexity of TARL is justified over simpler alternatives that achieve similar entropy reduction.

**Other Comments Or Suggestions:**

L426 test-time data instead of test-time date
L41 migrate -> mitigate
L346: Impletement -> Implementation

**Other Strengths And Weaknesses:**

Strengths:
- The implementation approach (only updating layer normalization parameters) is computationally efficient
- The method demonstrates compatibility with multiple offline RL algorithms
- The paper shows empirical improvements on standard benchmarks

The paper's primary weakness is its failure to validate the core hypothesis that training on low-entropy states transfers beneficial knowledge to other similar states in the environment. Without controlled experiments that explicitly track this knowledge transfer mechanism, it remains unclear whether the observed improvements stem from the claimed selective learning transfer or from simpler entropy reduction approaches.

**Questions For Authors:**

1. Your paper seems to rest on the hypothesis that selective training on low-entropy states enables beneficial knowledge transfer to other similar states. Could you explicitly articulate this hypothesis and provide direct evidence for this specific transfer mechanism rather than just overall performance improvements?

2. Have you conducted experiments comparing TARL to simpler alternatives that also reduce policy entropy:
   a) A policy that simply uses argmax action selection for states with entropy below a threshold?
   b) A policy that globally reduces temperature in the Boltzmann distribution?
   If so, what were the results? If not, why were these fundamental comparisons omitted?

3. Can you provide controlled experiments in simpler environments that explicitly demonstrate the hypothesized learning transfer from low-entropy states to related states? Ideally, this would include visualizations or state-by-state performance metrics showing how the selective training influences performance on states that weren't directly trained on.


Providing such evidence would significantly strengthen the paper and my rating.

**Relation To Broader Scientific Literature:**

The paper appropriately relates its contributions to existing offline RL approaches such as CQL, REM, and QR-DQN, as well as to existing test-time adaptation methods (e.g., TTT, TTT++, batchnorm adaptation).

**Theoretical Claims:**

N/A

---

> ### Author Rebuttal · Authors · 2025-04-01
>
> >Q1. Your paper seems to rest on the hypothesis that selective training on low-entropy states enables beneficial knowledge transfer to other similar states. The paper should provide ablation studies comparing selective entropy minimization against global entropy minimization.
>
> A1. Thank you for your valuable feedback. The rationale for focusing on low-entropy states operates on two levels:
>
> **1. Theoretical Motivation.** Low entropy directly reflects reduced action uncertainty. States with low-entropy correspond to more deterministic and reliable decision regions in the policy space. By prioritizing these low-entropy states during test-time adaptation, the policy can establish stable decision boundaries while avoiding OOD extrapolation.
>
> **2. More study on the advantage of selective training on low-entropy states.** The experiments were conducted in the walker2d-medium-v2 task to evaluate the hypothesis that selective training on low-entropy states improves TTA performance. The experimental setup included three conditions:
>
> * **Global Entropy Minimization**, where all available data were used for test-tiem adaptation.
> * **Low-entropy Selective Training**, where only samples with entropy below a predefined threshold were used for training.
> * **High-entropy Selective Training**, where only samples with entropy over a predefined threshold were used for training.
>
> Our results demonstrate that selective training on low-entropy states improves TTA performance more effectively than global entropy minimization. Meanwhile, if we only select high-entropy samples for tta, the performance actually becomes worse. This further indicates that selective training on low-entropy states enables beneficial knowledge transfer.
>
> Table C. Effectiveness of low-entropy selective training.
> |Global Entropy Minimization|Low-entropy Selective Training|High-entropy Selective Training|
> |-|-|-|
> |77.74|**82.95**|72.91|
>
>
> >Q2. Have you conducted experiments comparing TARL to simpler alternatives that also reduce policy entropy: a) argmax action selection, b) globally reduces temperature in the Boltzmann distribution? If so, what were the results? If not, why were these fundamental comparisons omitted?
>
> A2. Our study does not include comparative experiments between TARL and simpler entropy-reduction approaches like boltzmann or argmax action selection, primarily due to their incompatibility with test-time adaptation requirements.
>
> **1. Characteristics of Test-Time Adaptation**
>
> During the test-time phase, the model has already been trained. The primary goal is fine-tuning the model using test data to adapt to the actual environment. The key characteristics of this phase is that the model cannot access environmental reward signals or other forms of feedback. This means the model cannot rely on environment feedback to guide policy updates.
>
> **2. Limitations of Boltzmann and Argmax Policies**
>
> The Boltzmann policy is an action selection method that uses the softmax function to convert action values into a probability distribution. During training, the Boltzmann policy requires environment feedback to adjust the temperature parameter and action value function. The argmax policy is an action selection method that directly selects the action with the highest value. During training, the argmax policy needs environment feedback to evaluate and update the action value function.
> In the test-time phase, the lack of environment feedback makes them impossible to effectively update these parameters.
>
> **3. Advantages of TARL**
>
> TARL has no reliance on environment feedback. It uses entropy minimization as the objective function for policy updates during the test-time phase, effectively updating the policy. By minimizing entropy, TARL enhances the selection of high-confidence actions and reduces reliance on uncertain actions, better adapting to changes in the distribution of test data.
>
> >Q3. Can you provide controlled experiments in simpler environments that explicitly demonstrate the hypothesized learning transfer from low-entropy states to related states?
>
> A3. We have conducted ablation studies on the effect of low-entropy states selection in Section 4.7. $E_0$ controls the selection of samples for updating. Specifically, samples with entropy below this threshold are selected for updating. As $E_0$ decreases, the selected samples have lower entropy and fewer in number.
> When the entropy threshold increases, the performance of test-time adapted RL declines, which shows that it is necessary to filter some samples with relatively high confidence to update the offline policy to adapted the online environment.
>
> ---
> We sincerely hope our clarifications above have addressed your questions.

---

### Official Review · Reviewer_3AG2 · 2025-03-14

**Overall Recommendation:** 3

**Summary:**

This paper proposes TARL, a framework that minimizes action uncertainty at test time to mitigate distribution shift issues.

**Claims And Evidence:**

The authors conduct experiments on the D4RL and Atari benchmarks to validate the effectiveness of their framework.

**Essential References Not Discussed:**

I suggest adding more references to recent progress in this area.

**Experimental Designs Or Analyses:**

I suggest adding more experiments to further evaluate the effectiveness of the proposed framework. Specifically, implementing the TARL framework on top of other offline RL algorithms, such as IQL, could provide additional insights. Additionally, I recommend conducting experiments on other datasets included in the D4RL benchmark, such as the AntMaze environments and the replay and expert datasets in Gym environments, as this is a common practice in the field.

**Methods And Evaluation Criteria:**

Yes, the evaluation criteria adopted in this paper are standard in this field.

**Other Comments Or Suggestions:**

No further comments.

**Other Strengths And Weaknesses:**

Please refer to the Experimental Designs Or Analyses part.

**Questions For Authors:**

No further questions.

**Relation To Broader Scientific Literature:**

This work falls within the area of offline-to-online RL algorithms, which aim to mitigate distributional shift issues when encountering out-of-distribution state-action pairs.

**Theoretical Claims:**

Yes, although the paper does not present major theoretical results, I have verified the correctness of the proposed methods.

---

> ### Author Rebuttal · Authors · 2025-04-01
>
> > Q1. Implementing the TARL framework on top of other offline RL algorithms, such as IQL.
>
> A1. Thank you for your valuable feedback. The state-action out-of-distribution (OOD) issues stem from data distribution shifts between training and testing phases, rather than being specific to particular offline RL algorithm. Our proposed TARL method is specifically designed to mitigate OOD interference in offline RL algorithms, thereby significantly enhancing stability during testing phases.
>
> As a foundational algorithm in offline reinforcement learning, IQL [1] will suffer from this fundamental challenge during the testing phase. In Table A, we present a comparative analysis of IQL performance on Walker2d-medium-v2 environment and Walker2d-expert-v2 environment before and after TARL. Our approach achieving improvement in terms of average episode return.
>
> Table A. Effectiveness of our TARL applied to IQL.
> | Task Name | IQL | IQL (TARL) |
> | -------- | -------- | -------- |
> | walker2d-medium-v2     |   79.92   |   **82.43**   |
> |walker2d-expert-v2|110.31|**110.49** |
>
> [1] Offline reinforcement learning with implicit q-learning. ICLR 2022.
>
>
> > Q2. Conducting experiments on other datasets included in the D4RL benchmark, such as the AntMaze environments and the replay and expert datasets in Gym environments.
>
> A2. Accoding to your suggestion, we study more extensively across multiple benchmark datasets, including the AntMaze environments and the replay and expert datasets in Gym environment, to rigorously validate the effectiveness of our method. From Table B, our TARL achieves better performance than original RL algorithms, with particularly notable improvements in the complex environment Antmaze-medium-diverse-v0 and the expert-level task Hopper-expert-v2. These results further demonstrate the advantages of our method in handling data noise, distribution shift, and high-complexity tasks.
>
> Table B. Average episode return comparison of TARL against baseline methods on D4RL benchmarks over the 10 evaluations.
>
> Environment| Task Name | CQL | CQL (TARL) |
> | --------| -------- | -------- | -------- |
> | Mujoco| walker2d-expert-v2    |   113.25   |    **113.57**  |
> | Mujoco| walker2d-fully-replay-v2     |   95.31   |**97.99**
> |Mujoco|hopper-expert-v2|99.72|**113.34**
> | Antmaze| medium-play-v0   |   6   |**7**
> | Antmaze| medium-diverse-v0      |    2  |**5**
>
> ---
> We sincerely hope our clarifications above have addressed your concerns.

---

### Decision · Program_Chairs · 2025-05-01

**Decision:**

Accept (poster)

**Comment:**

This paper introduces a framework that mitigates distribution shift in offline RL by fine-tuning policies during deployment without requiring reward signals. Reviewers found the method novel, lightweight, and broadly applicable, with demonstrated improvements across Atari and D4RL benchmarks. However, concerns were raised about the lack of comparisons to simpler entropy-reduction strategies (e.g., global temperature reduction, argmax action selection) and related offline-to-online RL algorithms, as well as the marginal performance improvements in many tasks, raising questions about practical significance. While the rebuttal addressed some issues, such as additional experiments on IQL and ablation studies, the paper still lacks critical baselines and more direct evidence for its core hypotheses. The authors are encouraged to address these concerns in the final version.